# Salinity Stress Responses and Adaptation Mechanisms in Eukaryotic Green Microalgae

**DOI:** 10.3390/cells8121657

**Published:** 2019-12-17

**Authors:** Prateek Shetty, Margaret Mukami Gitau, Gergely Maróti

**Affiliations:** 1Institute of Plant Biology, Hungarian Academy of Sciences, Biological Research Centre, 6726 Szeged, Hungary; shettypr@brc.hu (P.S.); gitau.margaret@brc.hu (M.M.G.); 2Faculty of Water Sciences, National University of Public Service, 6500 Baja, Hungary

**Keywords:** high salt stress, green algae, adaptation, transcriptome, salinity, *Chlamydomonas*

## Abstract

High salinity is a challenging environmental stress for organisms to overcome. Unicellular photosynthetic microalgae are especially vulnerable as they have to grapple not only with ionic imbalance and osmotic stress but also with the generated reactive oxygen species (ROS) interfering with photosynthesis. This review attempts to compare and contrast mechanisms that algae, particularly the eukaryotic *Chlamydomonas* microalgae, exhibit in order to immediately respond to harsh conditions caused by high salinity. The review also collates adaptation mechanisms of freshwater algae strains under persistent high salt conditions. Understanding both short-term and long-term algal responses to high salinity is integral to further fundamental research in algal biology and biotechnology.

## 1. Introduction

Algae refer to a broad group of micro- and macroorganisms capable of oxygenic photosynthesis, yet show striking differences compared to land plants. The traditional and common definition of algae includes prokaryotic cyanobacteria and eukaryotic algae belonging to highly different phylogenetic clades. Eukaryotic microalgae are polyphyletic in origin and have an extensive ecological niche and evolutionary history. They are ubiquitous across most environments as primary producers, including extreme environments like soda pans and salt lakes. Soda pans are highly saline, shallow, and intermittent aquatic systems that are highly alkaline due to the high concentration of sodium and carbonate ions. Salt lakes represent another great example of hypersaline environments where green algae grow, and in some cases, thrive [1]. These environments have greater and often changing salt concentrations than that of seawater. Despite these extreme conditions, eukaryotic green algae remain primary producers in these ecosystems alongside cyanobacteria and euglenophytes [2]. Eukaryotic algae possess great plasticity and adaptability to most abiotic stresses. Some species belonging to the same genus are capable of growing in both fresh and saline water.

However, most freshwater strains show reduced growth and survival in high salinity environments and are generally unable to survive beyond a low threshold of salinity. In this review we will use the terminology of salt sensitive algae to refer to freshwater algae strains that have limited viability under high salt stress and salt tolerant algae to refer to marine or brackish algae strains that can easily cope with high salinity. Later on in the review, we will also be discussing the adaptation of salt sensitive strains to high salinity conditions and we will refer to these as salt adapted strains. We propose to use this unified terminology since multiple studies using different terminologies are being discussed and compared in this review.

Studying properties of salt tolerant algae species is a good first step in understanding how photosynthetic organisms cope with high salt. This is because unlike their freshwater counterparts, the machinery to tackle with salinity is engraved in the genome of marine algae as a result of centuries of evolutionary selection. In contrast, freshwater species have to devise mechanisms to cope with high salinity stress they are not accustomed to. This requires drastic changes in morphology and osmolyte concentrations in the short-term and accumulation of advantageous mutations in the long run. More than four decades of research has shaped our understanding of how microalgae respond, acclimatize, and grow under such challenging stress. However, there are few reviews tabulating all the multitude of changes that occur. Further, some of these changes are strongly species-specific. Thus, a comparison across different species can provide a more complete map of the different mechanisms used. Finally, a lot of commercially viable products are produced by microalgae to improve their survival under highly saline conditions. Exploitation of these mechanisms is particularly sought after in multiple fields. However, successful exploitation can only occur with a thorough understanding of these mechanisms and the propensity of the mechanisms to change under long-term adaptation.

Unfortunately, only a few species of salt tolerant *Chlamydomonas* are described. *Chlamydomonas pulsatilla* is a marine species isolated in Canada. Growth studies for this species have shown salinities of 10% artificial seawater (ASW) to be optimal for high growth rate, while reduced growth rate was observed at higher salinities. The growth rate at 200% ASW was 69% of the growth rate at 10% ASW [3]. This strain can utilize acetate as a carbon source, but it is incapable of utilizing nitrate or urea as nitrogen source. However, it can utilize several different amino acids for nitrogen [4]. Salt tolerant marine strains are valuable models to identify and characterize genes associated with salinity resistance. *Chlamydomonas W80* is a marine *Chlamydomonas* species, isolated from the Japanese coast [5]. A number of studies using cDNA libraries built from *C. W80* have successfully characterized a small group of genes that provided increased salinity tolerance in transformed *E. coli* cells. Genes from eukaryotic microalgae that are experimentally characterized as salt tolerance-related genes are shown in Table 1.

## 2. Effect of High Salt Concentration on Green Algae

There are multiple morphological and molecular changes that occur simultaneously that improve survival of salt sensitive algae under high salt stress. These are described in greater detail below.

### 2.1. Basic Morphological Changes

Stress caused by high salt concentration slows down cell division, reduces size, ceases motility, and triggers palmelloid formation in *Chlamydomonas* species [13,14,15,16]. Growth rate is immediately and directly impacted and can be easily detected. *Chlamydomonas reinhardtii* cells under high salt stress have lower growth rate compared to untreated cells. Untreated (control) salt sensitive algal cells reached an optical density (OD) of 1 at 750 nm in four days while it took the salt-treated cells six days to achieve the same OD. The cells were smaller in the dividing stage across all the salt concentrations [16]. Increasing salt concentrations negatively affected the growth of other freshwater algae species such as *Chlorella vulgaris*, *Chlorella salina*, *Chlorella emersonii* [17], and *Scenedesmus opoliensis* [18]. Figure 1 provides an overview of all the short term acclimatization responses that occur when *Chlamydomonas* is exposed to hypersaline conditions.

*Chlamydomonas* cells exposed to unfavorable conditions enter a temporary stage termed as “palmelloid”. Multiple structural changes occur in the palmelloid form including (i) loss of flagella, (ii) clustering of cells with a minimum of two cells per cluster, (iii) increased secretion of exopolysaccharide (EPS), (iv) cells surrounded by an EPS matrix sharing a common membrane, and (v) thickening of individual cell walls (Figure 1). The palmelloids increase steadily in number until 24 h after exposure to high salinity. A proteomic analysis identified expansin, Wall Stress-responsive Component (WSC) domain protein, pheophorin-C5, Vegetative Storage Protein rich protein (VSP4), and Cathepsin-Z-like proteins to be putatively linked to palmelloid formation [14]. The palmelloid structure of *C. reinhardtii* is highly interesting, the multicellular structures can be clearly visualized under microscope (Figure 2, Figure 3 and Figure 4, unpublished in-house data).

Under favorable conditions, the EPS matrix is degraded and the cells dissociate from their palmelloid form to enter the medium. Accumulation of a protein with a PAN/APPLE like domain in the post-stress spent media [14] backs up this phenomenon because such domains facilitate protein–carbohydrate interactions [19]. The same trend was observed for metalloproteinases Matrix Metallopeptidase 13 (MMP13) and Matrix Metallopeptidase 3 (MMP3), which are calcium-dependent endopeptidases involved in the degradation of EPS matrix such as mucilage [20].

Although the EPS matrix consists of different polysaccharides, the exact composition of EPS varies from species to species and environmental conditions. The production of EPS is energy intensive. However, the protection offered by the EPS matrix allows stressed cells to survive under adverse conditions by sequestering stressed cells away from the environment. Salt sensitive *Chlamydomonas* cells placed in a medium with 100–150 mM NaCl produced 3–6-fold more EPS compared to that in a medium without increased salt [14]. This implies that the increase in salinity modulated the increase of EPS production.

Algal genera such as *Dunaliella* and *Chlorella* do not form palmelloids and show different mechanisms to tackle with salt stress. *Chlorella* cells have a rigid cell wall, thus limiting its ability to change cell volume. Hence, osmoregulation through production of organic solutes and accumulation of inorganic ions are used to maintain osmotic homeostasis. Furthermore, cations in the media are bound within intracellular spaces reducing the osmotic activity [21]. In contrast, *Dunaliella* lacks a rigid cell wall which allows the cells to rapidly change the volume during high salinity stress by adjusting intracellular ion and glycerol concentration eventually restoring the cells turgor pressure [22]. A study on cell size changes in *D. salina* caused by high salt stress showed cell volumes continuously fluctuate for ten days, finally stabilizing at a cell size that was slightly larger compared to unstressed conditions [23].

### 2.2. Production of Osmoregulatory Solutes

Algal cells with rigid cell walls have limited ability to change cell volume and thus depend heavily on organic solutes for osmoregulation. These solutes, also termed as compatible solutes, are typically small organic molecules with neutral charge and low toxicity at high concentration [24]. Furthermore, compatible solutes at high concentrations allow for efficient functioning of different enzymes. Compatible solutes accumulate in the cytosol and balance the osmotic stress between the outside medium and the cytosol.

Glycerol is a good example of a common but effective compatible solute that is produced by most salt sensitive algal species under high saline stress. Glycerol is highly soluble and chemically inert, therefore it is non-toxic. It is an end-product metabolite and thus production and accumulation does not interfere with other metabolic pathways. Finally, glycerol production from starch is not expensive energetically and is not nitrogen-dependent [25].

In *Chlamydomonas HS-5*, glycerol accumulation corresponded to salt concentration with higher salt concentration causing higher glycerol content [26]. A similar effect was recorded for *C. reinhardtii* [27], *C. mexicana* [28], *Chlamydomonas sp. JSC4* [29], and *C. pulsatilla* [4]. A study in euryhaline *Chlorella autotrophica* showed similar increase in glycerol content in response to increasing salinity [21]. Glycerol plays a similar role in various *Dunaliella*, *Scenedesmus*, and *Micrasterias* species among other eukaryotic microalgae. Salt stressed *Dunaliella* cells accumulate massive amounts of glycerol and the level of intracellular glycerol was found to be proportional and osmotically equivalent to the external salt concentration [30]. The high concentration of glycerol also allows salt stressed *Dunaliella* cells to resume original cell volume even under extreme salt stress. Finally, starch degradation corresponded closely with glycerol accumulation in *C. pulsatilla* indicating that glycerol was synthesized through degradation of starch [31].

Glycerol biosynthesis (Figure 5) can occur through starch degradation or by using photosynthetic products [32]. Glucose produced through photosynthesis or starch hydrolysis is converted to fructose 1,6-bisphosphate and next to dihydroxyacetone-phosphate (DHAP), which is converted to glycerol-3-phosphate (G3P) by glycerol-3-phosphate dehydrogenase (G3PDH). G3PDH is the best studied enzyme in this biosynthetic pathway. G3PDH reduces DHAP to yield G3P, which is converted to glycerol by the action of a G3P phosphatase (GPP) or potentially through the reversible reaction of a glycerol kinase [33]. There are five isoforms of G3PDH enzyme in *C. reinhardtii.* Of these, GPDH_2_ and GPDH_3_ play integral roles in glycerol production and lipid synthesis [34].

While most strains produce glycerol and maintain it within the cell, some have the capacity to continuously leak significant quantities of glycerol into the surrounding environment. Releasing large quantities of photosynthetic products leads to an increase in photosynthetic rate in higher plants. *Arabidopsis* with extremely large sugar exporting veins can triple its photosynthetic capacity. Thus, it seems like there is an interplay between photosynthetic rate and the limited ability of the organism to use, export, or store the photosynthetically fixed carbon [35,36]. NADPH is continuously consumed to make up for released glycerol, thus, any inhibition in photosynthetic rate due to accumulation of NADPH and glycerol is prevented. Further, production and release of glycerol is linearly proportional to light intensity and exudation of glycerol led to increased growth rates in *C. reinhardtii* [37]. This is yet another pathway that algae can use to counter high salt stress and is observed especially in freshwater and marine *Chlamydomonas* cells [5,37] and in a single *Dunaliella* species [38].

Proline is another osmoregulatory solute that linearly increases in concentration with increase in salinity across higher plants and algae [39,40]. Like glycerol, it is low in molecular weight, neutrally charged and highly soluble. Exogenous application of proline reduces detrimental effects of high salinity by reducing accumulation of Na^+^ and Cl^−^ in *C. reinhardtii* [41] and plants [42]. Apart from proline, other amino acids such as lysine and leucine have been implicated in promoting *Chlamydomonas* growth under highly saline conditions [4]. However, not many studies are carried out on the specific roles of these amino acids in osmoregulation. Up-regulation of genes involved in proline synthesis has been recorded for *Picochlorum oklahomensis* [43] as well as *Picochlorum SE3* [44] during salt stress. Unlike in *Chlamydomonas sp.* and *D. salina*, where the major osmolyte is glycerol and starch degradation increases, proline is the principle osmolyte in *Picochlorum* species and starch synthesis is upregulated while starch degradation is limited [44,45,46].

Besides proline and glycerol, trehalose also has an established role in stabilization of proteins by increasing the transition temperature of proteins and as an osmoregulatory molecule [47]. High salt stress in *Chlamydomonas* [48], *Chlorella,* and *Scytonema* was shown to lead to an increased production of trehalose [49]. In maize, trehalose was observed to reduce the negative effects of high saline stress as an osmoprotectant [50]. Other polyols such as sorbitol and mannitol are also important in osmoregulation [44]. *Platymonas suecica* is a salt tolerant algae and shows a linear accumulates mannitol with increasing levels of salinity [51]. *Stichococcus chloranthus* and *Stichococcus bacillaris* both show an accumulation of sorbitol with increasing salinities [40,52]. Sorbitol and mannitol have not been detected in appreciable quantities in other algal species indicating that this might be a mechanism specific to these species.

### 2.3. Reorganization of Membrane Transport Proteins

Efficient uptake and export of ions through the cell membrane is another important strategy to cope with high salt stress by maintaining the intracellular ion balance. According to certain studies in plants, high concentration of Na^+^ can interfere with uptake of other cations, especially K^+^ [53,54]. Since K^+^ participates in a plethora of physiological functions in plants, it is integral to maintain the cytosolic K^+^/Na^+^ ratio under highly saline conditions.

Upregulation of membrane transport proteins can confer tolerance to high salinity in halotolerant algae species by actively transporting K^+^ ions through membrane transport proteins. Salt adapted mutants of *C. reinhardtii* showed enhanced expression of multiple membrane transport proteins [55]. Genes for K^+^ ion transport were significantly upregulated when salt sensitive *C. reinhardtii* cells were under salt stress [48], possibly compensating the disruption in K^+^ uptake caused due to high concentration of Na^+^ ions. 

Intracellular ion content data revealed that *C. pulsatilla* had a remarkable capacity to increase sodium and chloride, and, to a lesser extent, potassium and magnesium in response to increased salinity, although accumulation of glycerol remained the principal approach to cope with salinity stress [3]. Further, salt tolerant strains of *Dunaliella* maintain intracellular Na^+^ concentrations below that of the outside medium [56]. The Na^+^/H^+^ antiporter catalyzes influx of Na^+^, followed by an export of Na^+^ through the Na^+^-ATPase and plasma membrane electron transport system. Thus, this Na^+^ extrusion in *Dunaliella* is an evolved, adaptive mechanism in hypersaline environments to maintain Na^+^ concentrations within cells. Thus, it seems that unlike marine or hypersaline algae, freshwater or salt sensitive strains are unable to maintain the intracellular ion concentration by preventing uptake of Na^+^, by accumulating K^+^ or pumping out Na^+^ ions to restore the cells to their normal or nearly normal homeostatic state.

Apart from membrane transport proteins, the amount of two plasma membrane proteins P150 and P60 were greatly elevated with increasing salt concentration in *D. salina* [57,58]. Moreover, immediately after a drastic hyperosmotic shock, the induction of these proteins coincided with increased growth suggesting their role in conferring salt tolerance. P150 is a 150 kD transferrin-like plasma membrane protein involved in iron uptake, thus it helps the cells overcome any possible limitation of iron availability under high salinity [58]. However, this protein was not detected in other algae and the most similar protein is found only in animals. We also carried out a basic blast analysis using the cDNA sequence of P150 (accession number: AAF72064.1, data not shown) and found no matches in any other algal species, indicating that this protein and mechanism is fully specific to *D. salina*. However, proteins related to iron uptake were upregulated in salt adapted *C. reinhardtii CC-503* indicating that iron uptake under high salt stress might be limiting to *C. reinhardtii* as well [55]. P60 is a novel type of carbonic anhydrase and is potentially involved in CO_2_ fixation within cells growing under hypersaline conditions [57]. The upregulation of three types of carbonic anhydrases was observed in salt tolerant *Chlorella pyrenoidosa 820* [59]. However, no such relationships have been reported in any type of *Chlamydomonas* species, despite the presence of 12 different carbonic anhydrases in *Chlamydomonas reinhardtii* [60].

### 2.4. Lipid Accumulation

Most green algae under salt stress show characteristic accumulation of lipids. In fact, this response to high salinity has been studied for the production of biofuel [28]. Lipids are produced as high energy storage compounds, synthesized when algal cells are under unfavorable environmental conditions. Once the algal cells are moved to optimal conditions, these lipid molecules are utilized. Lipids are synthesized by assimilating carbon from three different, independently regulated metabolic pathways. The first is direct incorporation of photosynthetically fixed CO_2_ into fatty acids within the chloroplast. The second involves degradation of starch. The third is through degradation of polar lipids [61]. However, no study has quantified the contribution of these pathways under normal conditions. 

*Chlamydomonas sp. JSC4* is a salt tolerant strain isolated from a marine environment and shows high lipid accumulation under high salt stress. This strain was utilized to identify the mechanism of lipid biosynthesis under high salt stress [29,62]. *Chlamydomonas sp. JSC4* shows a highly specific switch from starch synthesis to lipid synthesis under salt stress [29]. Presumably, the brackish water heritage of *Chlamydomonas* sp. *JSC4* selected for and adapted this algal strain to cope with higher salinity as the strain moved from brackish water to marine conditions. The switch from starch to lipid synthesis ensures the maintenance of energy reserves, thereby ensuring long-term survival. Degradation of accumulated starch leads to the production of an extremely important precursor of lipid production; G3P [62]. G3P is also an extremely important precursor for glycerol synthesis in eukaryotic green algae, as discussed earlier. Finally, all the glycerol accumulated can also be converted back to G3P through glycerol kinase [63]. However, no study has investigated this possibility in green algae so far.

However, it is important to note that lipid accumulation is not a strategy that is identified in all algae. This is especially true for salt adapted algae. A few studies have explored the evolution of salt resistance in green algae under progressively increasing saline conditions. This selection strategy was used to generate salt adapted mutants from salt sensitive *C. reinhardtii* [64], salt sensitive *Chlorella sp. AE10* [65], and a salt tolerant marine *Chlamydomonas JSC4* [66]. Transcriptome analyses of all the adapted mutants showed decreased lipid accumulation. Furthermore, the salt adapted (originally salt tolerant) *Chlamydomonas JSC4* also lost its switching capability from starch synthesis to lipid synthesis. There was only one adaptation experiment where adaptation to high salinity also led to increased lipid accumulation. A salt tolerant *Schizochytrium sp.* was adapted to increased salt stress, and the adapted mutant displayed increased lipid accumulation [67]. These studies clearly show that salt adaptation can occur in multiple different, species specific ways.

### 2.5. Impact of High Salt Stress on Photosynthesis

Most photosynthetic organisms show a significant decrease in photosynthetic activity under high salt stress. This reduction in photosynthetic activity can be attributed to deficiency in different cations, production of reactive oxygen species (ROS) and osmotic stress, which interfere with various biochemical and physiological processes [68]. Pigment analysis in *C. reinhardtii* has demonstrated that the photosystem I (PSI) light harvesting complexes (LHCs) are damaged by reactive oxygen species (ROS) at high salt conditions, and photosystem II (PSII) proteins involved in oxygen evolution are also impaired [16,69]. Salinity stress seems to divert the resources from PSII’s D1 protein turnover to the intensive process of maintaining cell homeostasis [16]. Transcriptome studies in *C. reinhardtii* have shown impairment of photosynthesis, with several PSI LHC genes being significantly down-regulated, e.g., PSI light harvesting complex genes LHCA2, LHCA3, and LHCA5. Furthermore, the levels of most of the chloroplast encoded transcripts (e.g., *psaA*, *B*, *C*, *J*, *M*) in PSI were relatively unchanged while the nuclear genes (e.g., *psaD*, *E*, *G*, *F*, *H*) were down-regulated under highly saline conditions [48].

Several transcriptomic studies have led to the implication that salt stressed cells upregulate different genes to resist harsh conditions and loss in photosynthetic activity. Transcriptome study of salt stressed *C. reinhardtii* cells showed significant upregulation of genes involved in eliminating ROS including plastid Fe superoxide dismutase 1 (SOD), thioredoxins, glutathione transferase, and heat shock factor binding proteins [64]. Pigments such as carotenoids play important functional roles as antioxidants. Carotenoids are lipid soluble antioxidants and are mainly located inside the chloroplast envelope and protects the LHC against ROS induced damage. *Dunaliella sp.* are good examples of algal strains that can produce high quantities of carotenoids in response to salt stress and have been industrially exploited for carotenoid production [70]. Another study on salt sensitive *C. reinhardtii* and *C. vulgaris* showed increased carotenoid production at moderate levels of salt stress (0.05 M–0.15 M).

Other species cope with salt stress in a different fashion. The cells of salt tolerant *D. salina* increase photosynthetic activity by significantly increasing Chlorophyll-a (Chl a) content in response to salt stress [17]. This is consistent with another study in salt tolerant *D. salina* that showed Chl a/b ratio slightly increased with increased salt concentration (3 M NaCl) [71]. In salt sensitive *C. vulgaris*, chlorophyll content was increased at lower salt concentrations (0.2 M) but reduced at higher salt concentrations (0.3 M–0.4 M) which slowed growth [72]. A slight increase in photosynthetic activity is only seen under moderate levels of salt stress and is presumably done in order to enhance energy generation to produce energetically expensive molecules that can protect salt stressed cells (e.g., EPS and pigments), or to drive energetically expensive Na^+^ exclusion or to increase amount of storage molecules like lipids. Thus, salt sensitive strains and salt tolerant strains seem to have different mechanisms of tackling with salt stress. The increase in photosynthetic activity for salt tolerant strains at moderately increased salinities is likely a regulatory response since these strains often need to deal with variable levels of salinity.

### 2.6. Glycolysis

Glycolysis is considered to play an important role in plant development and adaptation to multiple abiotic stresses, such as cold, salt, and drought [73,74]. In a transcriptome study of salt stressed *C. reinhardtii,* a significantly increased expression of genes participating in the metabolism of carbohydrates, such as starch, sucrose, soluble sugar, and glucose was observed. Furthermore, salt stressed cells showed an upregulation of 31 genes involved in glycolytic processes, including plastidic pyruvate kinase PKP-ALPHA and PKP-BETA1 [48]. This is consistent with the results observed in salt adapted *Chlorella sp. S30* [66] and in plant studies [75] where high salt stress significantly increased the main carbohydrate contents. Carbohydrates are involved not only in osmotic adjustment, but also can be used as protective agents for homeostasis as osmoprotective solutes as previously described.

High saline stress negatively impacts photosynthetic efficiency in sensitive cells, which often inhibits transport of carbohydrates leading to an accumulation of excess starch or sucrose. *C. reinhardtii* enhanced glycolysis to decrease carbohydrate accumulation in cells, which would promote the respiratory metabolism and mitochondrial electron transport. Thus reducing the effects of ionic toxicity and osmotic stress caused by excess salt [48]. Instead the carbon from glycolysis is used in lipid production and ATP generated could be directed into energy intensive mechanisms described above to improve survival under salt stress. Further, as mentioned in Section 2.2, glycolysis is the major pathway that produces osmoregulatory solutes like glycerol. Glycolysis products can also be utilized in lipid production. Thus, regulation of this pathway is highly important under high salt stress (Figure 5).

### 2.7. Role of Acetate

Studies have demonstrated the use of acetate in the medium as an alternative source of energy to compensate for the lowered efficiency in photosynthesis [76]. Transcriptome studies of high salt stressed *C. reinhardtii* cells show an upregulation of acetyl-CoA synthetase encoding gene, which combines acetate and CoA to form acetyl-CoA [64]. Upregulation of acetate metabolism related genes like acetyl-CoA carboxylase and dehydrogenase a were also observed in transcriptomic studies of the halophyte *Prymnesium parvum*, along with homologs to Type I and Type III polyketide synthases [77]. Salt adapted mutants of *C. reinhardtii* showed an improved growth, when salt stressed cells were growing with acetate. More importantly, a greater reduction in growth was observed when the mutants were grown without acetate in salt free media. Further there was significant upregulation of TCA related genes [64]. These results suggest that acetate is introduced into energy generation pathways such as TCA cycle in salt stressed algal cells to compensate for reduction in photosynthesis.

## 3. Development of Salt Tolerant Freshwater Algae

Directed experimental evolution by selecting for survivors under a progressively increasing stress has been successfully applied to generate salt adapted algae from salt sensitive progenitor cells. Salt adapted algae strains were selected through multiple generations of cultivation under saline stress. Salt tolerance was developed by the 1255th cycle and by the 46th cycle in *C. reinhardtii* [64] and *Chlorella sp. AE10* [65], respectively. Transcriptomic data revealed that responses to salt stress were different in salt sensitive progenitor and salt adapted evolved mutants. The progenitor salt sensitive *C. reinhardtii* cells displayed a reduction in photosynthesis, upregulation of glycerophospholipid signaling, and upregulation of transcription and translation machinery. In contrast, salt adapted *C. reinhardtii* cells showed downregulation of genes responsible for lipid accumulation and the transcription/translation mechanisms. Similar results were observed in salt adapted *Chlorella sp*. *AE10* strains [65].

Directed experimental evolution coupled with random mutagenesis through irradiation has been applied in *Chlamydomonas sp. JSC4* to improve biomass production under saline conditions [66]. Irradiation using heavy ion beam was performed to induce mutations before long-term and continuous cultivation. Heavy ion beams have been observed to be effective mutagens for application in mutational breeding because they induce several types of mutations such as single and short nucleotide alterations as well as large insertions/deletions in the genomic DNA [78]. Nevertheless, spontaneous mutations resulting from long-term cultivation also proved to be efficient, however, not to the same extent as the irradiation-induced mutagenesis. No increase in biomass or lipid accumulation was identified in these salt adapted mutants [66]. Taken together, the lack of lipid accumulation in salt adapted algae is a highly interesting and important result, especially for those studies looking to utilize salt stress induced algal lipid accumulation for biofuel purposes. 

Transformation of salt tolerance conferring genes is a feasible way of enhancing freshwater strains ability to cope with high salt stress. While it has been successfully applied for cyanobacteria where the stress tolerance genes were simply overexpressed [79], limited studies have carried out transformation using specific genes to confer salt tolerance in eukaryotic algae (Table 1).

Other methods such as genome shuffling through selective sexual breeding have also been successfully used to develop salt adapted algae strains. This is a promising alternative to quickly generate salt adapted algae. Unfortunately, so far only a single study has utilized this approach to generate a salt adapted *C. reinhardtii* strain [80]. By comparing the genomes of the newly evolved salt adapted strain to the genome of its parental salt sensitive strain, genes that might confer protection under salt stress could be identified. Sexual breeding pathways were identified in various *Scenedesmus* [81], *Chlamydomonas* [82], and *Dunaliella* species [83]. Coupling random mutagenesis approaches along with sexual breeding of selected mutants is a promising combination method to develop salt adapted strains in multiple, biotechnologically important algal strains. It is mainly the result of selective breeding strategies that we currently have a diverse pool of agricultural crops with tolerance to different abiotic stresses. These techniques could be applied to algal species as well in order to generate economically viable algal variants.

## 4. Perspectives of Salt Stress Utilization in Biotechnological Applications

Salt stress can be utilized to generate a plethora of valuable products. Multiple studies have been carried out on the exploitation of high salinity condition as a methodology to improve algal lipid production. Overall, salt stress promotes lipid and β-carotene production which is a phenomenon that can be exploited and commercialized. Such phenomenon was observed for β-carotene production in *D. salina* [70,84]. It was observed that pairing nitrogen deficiency and high salt stress could be a feasible way of enhancing glycerol production using microalgae [85].

However, it is important to differentiate between processes that occur in salt sensitive algae, salt adapted mutants and salt tolerant strains. Studies have shown that gene expression patterns change as algae adapt to high salt stress. Furthermore, gene expression patterns under high salt stress are strikingly different in salt tolerant algae strains and salt adapted mutants. As previously mentioned, salt adapted mutants of *C. reinhardtii* do not show the characteristic lipid accumulation response to high salt stress as observed in salt sensitive and in salt tolerant strains. This observation indicates that there are various adaptation pathways to high salt stress. Further, adaptation to salt stress is species specific and depends greatly on the life history of the strain. In many cases salt adapted algae might not be suitable candidates for biotechnological applications such as biodiesel production, because adaptation of a salt sensitive, lipid producing strain might just produce a mutant that is salt tolerant but may lose its lipid accumulation capacity. Thus, we propose to utilize salt tolerant strains coupled with sexual breeding approaches to improve lipid production rather than to harness adaptation of salt sensitive strains. Sexual breeding approaches might be better suited to select and conserve pathways of algal lipid accumulation compared to non-specific mutational approaches. A lot of commercial algae cultivations prefer indoor and closed cultivation techniques to avoid fatal contaminations, thereby strongly increasing the cost of cultivation and technology required. The use of salinity as a crop protection mechanism is also feasible. It has been applied as an effective protection method for reducing the microbial contaminants in *Picochlorum (Nannochloris) atomus* freshwater green algae cultures [86]. Reliance on salinity as a control measure allows the harvesting of high quality biomass, which reduces economic losses pertaining to culture re-establishment and end-product loss. *Picochlorum SE3*, isolated from a shallow mesophilic brackish-water lagoon, can tolerate a hypervariable environment making it a suitable candidate for large-scale open-pond cultivation [44]. Similarly, various mutant eukaryotic green algae with evolved salt tolerance features [65,66] can be grown in saline media with no reduction in biomass, thus reducing the dependency on freshwater sources. Hence, high quality algae biomass with limited contamination can be cultivated using either salt tolerant and salt adapted algal strains.

## 5. Conclusions

More than four decades of research has drastically enriched our understanding of the various mechanisms used by microalgae to cope with salt stress. Initial studies exploring accumulation of osmolyte concentration provided us with an understanding of what kinds of osmolytes are accumulated (such as glycerol) and why these osmolytes are important. Physiological changes such as *C. reinhardtii* cells entering the palmelloid stage or increased production of EPS and cell size changes are all integral to further improve osmoregulation. Contemporary differential expression studies have begun to probe the pathways that are differentially regulated under optimal and high salt stressed conditions. Most algal cells show a characteristic accumulation of lipids under salt stress. Lipids act as storage reserves for salt stressed cells and can be immediately degraded when stressed cells are introduced to optimal conditions. These studies have also improved our understanding of how products from pathways such as glycolysis, Kennedy pathway, and Calvin cycle are all used to improve both osmolyte production and reserves of storage molecules, mostly lipids (Figure 5). Studying diverse algal species such as *Prymnesium parvum* [77] and salt adapted mutants of *C. reinhardtii* [64] showed that algae can adapt to loss in photosynthetic efficiency by taking up acetate to improve energy generation and carbon assimilation. Finally, differential expression analysis studies have also enhanced our understanding of how transport proteins are involved in osmoregulation.

However, there is much uncovered ground in understanding how algae signal each other to begin the cascade of pathways that will lead to improved survival. One such example is the study of volatile organic compounds (VOCs) that are released when algal cells are under stress. A preliminary study has shown that VOCs like hexanal and longifolene are released from salt stressed *C. reinhardtii* cells. When these compounds are collected and exposed to algal cells growing under optimal conditions, a reduction in cell density, a slight increase in chlorophyll and an increase in antioxidant enzyme activity was observed [87]. Presumably, the VOCs signal unstressed cells to prepare for an oncoming stressful condition. Thus, future studies need to concentrate on signaling pathways that begin the cascade of physiological and metabolic changes that are described in this review. Further, while there have been a handful of studies on directed experimental evolution to improve salt tolerance, none of these have characterized the changes that are occurring in the genomes of these adapted organisms. Such studies coupled with next generation sequencing strategies and directed experimental evolution approaches will continue to further improve and deepen our understanding of how algae respond and adapt to stressful conditions.

## Figures and Tables

**Figure 1 cells-08-01657-f001:**
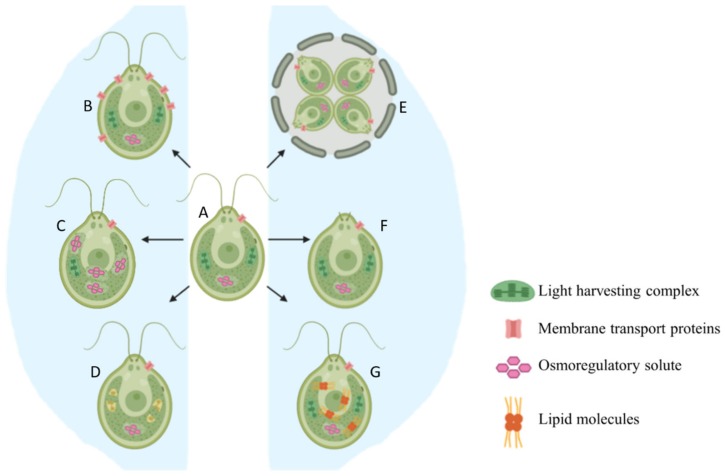
Conceptual image detailing the morphological changes that occur when normal cell (A) is exposed to saline conditions (B–G). (**A**) A *C. reinhardtii* cell under no stress, (**B**) Upregulation of membrane transport proteins, (**C**) Accumulation of osmoregulatory solutes, (**D**) Degradation of light harvesting complexes, (**E**) Palmelloid formation, (**F**) Flagellar loss and reduction of motility, (**G**) Accumulation of lipids.

**Figure 2 cells-08-01657-f002:**
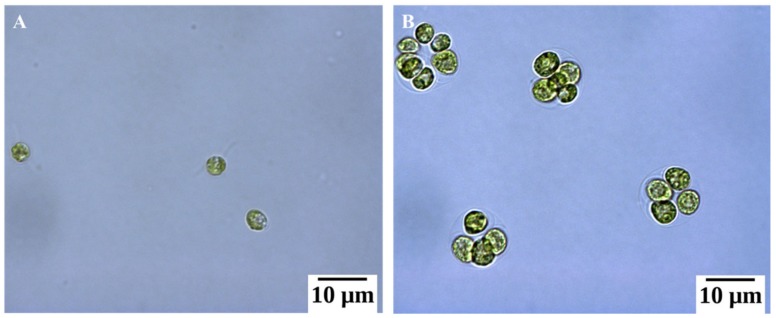
Optical microscopy images of *Chlamydomonas reinhardtii* cc124 under normal condition (**A**) and under salt stressed (150 mM NaCl) condition (**B**) (unpublished in-house data).

**Figure 3 cells-08-01657-f003:**
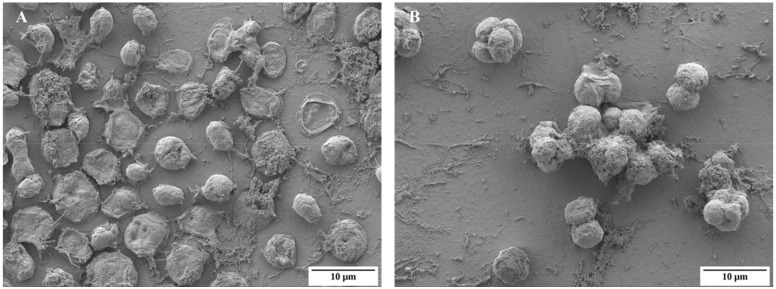
Electron microscopy images of *Chlamydomonas reinhardtii* cc124 under normal condition (**A**) and under salt stressed (150 mM NaCl) condition (**B**) (unpublished in-house data).

**Figure 4 cells-08-01657-f004:**
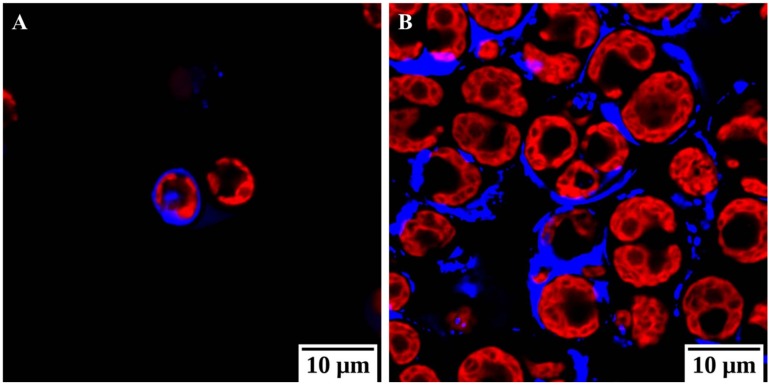
Confocal microscopy images of *Chlamydomonas reinhardtii* cc124 under normal condition (**A**) and under salt stressed (150 mM NaCl) condition (**B**). Calcofluor white stains cellulose and chitin and is blue in color while photosystem II was excited and visualized in red. These images show an increase in polysaccharides as an integral event of the palmelloid formation (unpublished in-house data).

**Figure 5 cells-08-01657-f005:**
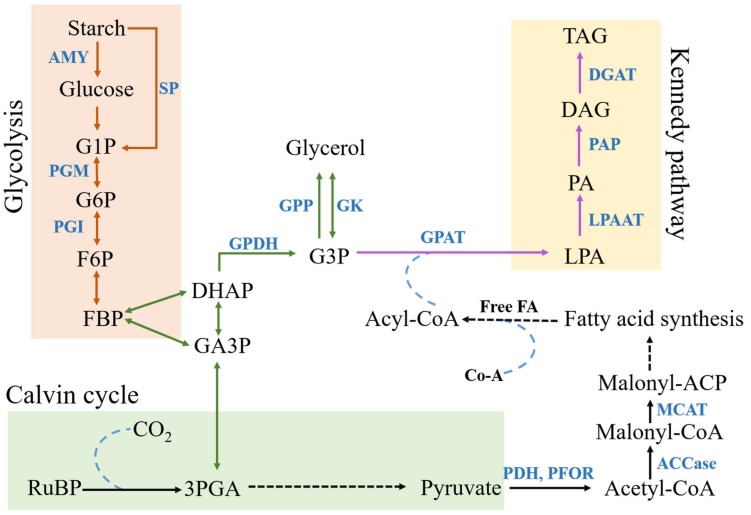
Schematic diagram of Glycerol and TAG synthesis. Metabolites: G1P- glucose 1-phosphate, G6P- glucose 6-phosphate, F6P- fructose 6-phosphate, FBP-, DHAP- dihydroxyacetone phosphate, GA3P- glyceraldehyde 3- phosphate, G3P- glycerol 3-phosphate, RuBP- ribulose 1,5-bisphosphate carboxylase/oxygenase, 3PGA- 3-phosphoglycerate, LPA-, PA-, DAG-, TAG-. Enzymes: AMY- α-amylase, SP- starch phosphorylase, PGM- phosphoglucomutase, PGI- phosphoglucoisomerase, GPDH- glycerol 3-phosphate dehydrogenase, GPAT- glycerol-3-phosphate acyltransferase, LPAAT- lysophosphatidic acid acyltransferase, PAP- phosphatidate phosphatase, DGAT- Diacylglycerol acyltransferase, MCAT- Malonyl CoA-acyl carrier protein transacylase, ACCase- AcCoA carboxylase, PDH- pyruvate dehydrogenase, PFOR- pyruvate-ferredoxin oxidoreductase, GK- glycerol kinase, GPP- glycerol-3-phosphate phosphatase. Others: Free FA- free fatty acids, Co-A- Coenzyme A.

**Table 1 cells-08-01657-t001:** Genes with experimentally observed functional response to salt stress.

Gene Name	Organism From	Organism Transformed	Effect Observed	Reference
Cyclophilins (CYP) PsCYP1	*Pyropia seriata (marine red algae)*	*Chlamydomonas sp.*	Heat and salt tolerance	[6]
Photosynthetic ferredoxin (PETF) and ferredoxin-5 gene (FDX5)	*Chlamydomonas sp.*	*Chlamydomonas sp.*	Salt tolerance	[7]
Gene with unknown function	*Chlamydomonas W80*	*E. coli*	Salt and cadmium tolerance	[8]
Anti-stress genes	*Chlamydomonas W80*	*E. coli*	Salt stress protection	[9]
Breast basic conserved gene (bbc1)	*Chlamydomonas W80*	*E. coli*	Salt stress protection	[10]
Glutathione peroxidase	*Chlamydomonas W80*	Tobacco plant	Salt stress protection	[11]
*Group-3 late embryogenesis abundant protein gene (cw80lea3)*	*Chlamydomonas W80*	*Synechococcus* PCC7942	Salt and cold stress protection	[12]

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
