# Peer review of "Salinity Stress Responses and Adaptation Mechanisms in Eukaryotic Green Microalgae"

_cells, 2019, doi:10.3390/cells8121657_

Round 1

Reviewer 1 Report

I’ve found many drawbacks in this manuscript.

The main problem in the review is that it presents a mix or collection of some facts and results from the available literature. For the review to be interesting, data from the literature should be reviewed and rethought (reinterpreted). In addition, it would be worth discussing the possible causes of contradictory results in the literature

Unfortunately, the authors admit the “copy and paste” approach and inserted only slightly changed fragments of the original works. Furthemore, there is a confusion of references in the review. The cited papers are often not relevant to the topic under discussion.

I have some doubts also concerning the relevance of Sections 3 and 4; they should be at least reduced. A conclusion that combines the data on the effect of salinity on the processes occurring in algae cells would be preferable.

The style of the manuscript needs revising.

In addition, I have comments, provided in detailed below.

Title

I recommend revising the title with changing “Salinity” on “Salinity stress” (or something like that) and also specifying the genus Chlamydomonas.

Abstract

Revise Abstract in order to minimize introduction in it (now it takes 8 lines L. 10-17), and focus on the main topic of the review.

The statement “Salinity is an adverse stress and a challenging barrier for algae to adapt to” is not correct, since salinity is one of the major factors controlling the growth rate of microalgae. The microalgae species grow at different salinities. Some species are suitable to marine condition, whereas others show optimum growth at lower salinity. Only a high deviation from the optimal salinity is a stress factor but not “salinity” itself. Really, high salinity stress is the most severe environmental stress. This understanding is also important for the correct title.

“Salinity in marine environments can vary greatly, with high salinity concentration at the tropics to low salinity concentration at the poles”. It is also incorrect because it is not so simply.

What does “Changes in tidal actions” mean?

Introduction

I recommend following the main topic and journal scope, while now the first part of Introduction considers ecology and the second part contains a short description of the impact of salinity on Chlamydomonas. It is necessary to identify the problems and explain the purpose of the review.

137-141. Figure 1. Is this your original data or was published earlier? Specify. 188-191. Who hypothesized? Needs a reference. 191-194. Needs a reference Burch et al., 2015. 197-198. Needs a reference.

You allotted L. 171-187 to glycerol, and only one L. 214 to sorbitol. Why?

229. “Chlorella sp. S30 variant”. May be strain?

As to Section 2.4. This subsection begins with the words “Most green algae under salt stress show characteristic accumulation of lipids”, but you limited the description of accumulation of lysophosphatidic acid. Nevertheless, the importance of lipids in many adaptation processes, photosynthesis, membrane permeability, etc. is obvious.

229–232. “In salt adapted Chlorella sp. S30 variant, the ABC transporter protein was upregulated by more than 10 fold.” What is the term "variant" used? The authors [48] named it strain. To be more correct, “the salt stress tolerance mechanism was up-regulations of some genes related to ABC transporters” [48]; moreover, not only ABC transporters, but also antioxidant enzymes, CO2 fixation, amino acid biosynthesis, and central carbon metabolism [48]. L. 233. Specify what the proteins P150 and P60 are responsible for. 236-239. D. salina is not strains, it is a species. Provide reference. Check the wording. 255-260. The species C. reinhardtii is mistakenly mentioned. It should be Chlamydomonas moewusii according references 51 and 52. L. 266-271. This fragment is entirely taken from Wang et al., 2018: “Further, analysis of glycerophospholipid metabolism pathways showed that the alga cells had significant up-regulation of FAD (flavin adenine dinucleotide)-dependent oxidoreductase family protein (Cluster-2749.52046; L2fc = 2.695) that involves storing lipid catabolism and glycerol assimilation, and in glycerol-3-phosphate shuttle, which transports reduced power from cytosol to mitochondrion [8]”. The exception is the last and important conclusion of Wang et al., 2018: “This suggests that the intracellular glycerol pool in reinhardtii cells likely increased as a response to salt stress, similar to what has been shown for the green alga Dunaliella tertiolecta”. 272. Reference 48 does not concern C. reinhardtii. Correct. L. 255 The statement “Most green algae under salt stress show characteristic accumulation of lipids” contradicts 276-278 “study involving multiple independent cell lines of C. reinhardtii shows similar response with salt adaptation negatively affecting lipid accumulation [55].” L. 275-276. “These include a phospholipid/glycerol acyltransferase putatively involved in PA synthesis.” It is difficult to agree with this statement, because, according to Arisz et al., 2003, “In response to various environmental stress conditions, plants rapidly form the intracellular lipid second messenger phosphatidic acid (PA). It can be generated by two independent signalling pathways via phospholipase D (PLD) and via phospholipase C (PLC) in combination with diacylglycerol kinase (DGK).” 286-287. “Energy metabolism pathways such as photosynthesis [48,54,56] and photorespiration [48] are severely affected.” What does it mean? 295-297. “photosynthesis is impaired during salt stress” This statement is not consistent with the statement below L. 307-311 and L. 317-318 “To cope with high salinity, the cells of marine D. salina increase photosynthetic activity”. Similarly, “significant increase of Chlorophyll-a (Chl a) in response to increasing salt concentration” L.318-319 is not consistent with “In C. vulgaris, chlorophyll content was increased at lower concentrations but reduced at higher concentrations” (L. 313-315). It should be explained and discussed. 297-300. Nothing of the mentioned here is discussed in Li et al., 2018 [48]. Check carefully. 307-311. Nothing of the mentioned here discussed in Mendoza et al., 1999 [59]. Check carefully the references. 340. What does “sensitive cells” mean? 340-345. Provide a reference.

Author Response

Please find enclosed our answers to the questions.

Reviewer 2 Report

This review presents an overview of effect of salinity stress on green algae, response of algae to salinity stress, and applications of salt stress in algal culture/biomass production. This manuscript is well-edited and convincing.

Therefore, I recommend this manuscript for publication.

It is better to clearly explain the objectives of this review paper at the end of Introduction part. And, please add the sufficient and appropriate citations at L333, L339, and L340-345.

Author Response

Thank you for the positive opinion. The manuscript was substantially revised, we have added a number of new section in various parts of the manuscript including the introduction part as well. You can follow the major changes highlighted in red. Also, all references were strictly reviewed, some of the references have been corrected and changed to even more relevant ones.

Reviewer 3 Report

I have read carefully the manuscript cells-619 282 entitled “Salinity responses in green microalgae: A review ”.The authors of this work present here a paper they wrote aboutthe effect of salinity for microalgae belonging to green lineage.”.

Hereafter you will find the major comments about this article.

My principal remark will concern the main substance of this article that is clearly preliminary and need more investigation to assess a real review paper stage. An intensive work is mandatory before envisaging to resubmit such an article. For this main reason, I am in regret to announce that I cannot recommend this manuscript for publication in the journal of cells even into a revised form.

Author Response

Thank you for your comments and for reading our manuscript. The manuscript was substantially revised, new sections were added, other whole sections were entirely re-written and merged. Since the very high number of changes we decided to prepare a version, where you can easily follow the major changes which are highlighted in red (the numerous further minor changes are not highlighted).

We think that by clearly discriminating between salt adapted (originally sensitive) and salt tolerant (mostly marine) algae in this revised version, we can propose explanations for the existing discrepancies in the literature concerning salt stress in microalgae.

Reviewer 4 Report

The paper of Shetty et al. is an important review about the multi-level response of green algae to salt stress, dealing with what we know so far and what are the available strategies to use and modify this response. This review is very important for all members of the community devoted to microalgal research and bring together a lot of literature providing a very nice comprehensive overview. It is nicely written, and will provide a nice literature for interested students and scientists. Having said these, the review could be improved in the following several aspects:

Major remarks:

1.       In the introduction (p. 2, line 56), the authors introduce the information on the glycerol content which, I have the impression, later is used for both: neutral lipid (oil, in the form of diacyl- and triacylglycerol, DAG and TAG) and osmolyte (glycerol). This can be quite confusing for the readers, especially working on microalgal lipids.  Moreover, the aspect of oil accumulation in response to salt stress is underrepresented in this review, meanwhile it could be quite interesting for the audience focused on research related to microalgae based biofuels (mainly TAG). I strongly suggest to include additional chapter on this aspect of salt stress response in green algae as there is quite a few references regarding this topic, including the most recent ones:

·         Hounslow E, Kapoore RV, Vaidyanathan S, Gilmour DJ, Wright PC. The Search for a Lipid Trigger: The Effect of Salt Stress on the Lipid Profile of the Model Microalgal Species Chlamydomonas reinhardtii for Biofuels Production. Curr Biotechnol. 2016;5(4):305–313. doi:10.2174/2211550105666160322234434

·         Atikij T, Syaputri Y, Iwahashi H, et al. Enhanced Lipid Production and Molecular Dynamics under Salinity Stress in Green Microalga Chlamydomonas reinhardtii (137C). Mar Drugs. 2019;17(8):484. Published 2019 Aug 20. doi:10.3390/md17080484

·         Kakarla R, Choi JW, Yun JH, Kim BH, Heo J, Lee S, Cho DH, Ramanan R, Kim HS. (2018). Application of high-salinity stress for enhancing the lipid productivity of Chlorella sorokiniana HS1 in a two-phase process. J Microbiol. 56: 56-64.

·         Ho SH, Nakanishi A, Kato Y, Yamasaki H, Chang JS, Misawa N, Hirose Y, Minagawa J, Hasunuma T, Kondo A. (2017). Dynamic metabolic profiling together with transcription analysis reveals salinity-induced starch-to-lipid biosynthesis in alga Chlamydomonas sp. JSC4. Sci Rep. 7:45471.

This separate chapter on storage lipids would fix some serious shortcomings on the topic in the present version of the manuscript, like p.2, line 65-67 or p.9, lines 276-278.

2.       Some references are missing in the text, like [71], therefore this manuscript need a thorough revising to be sure the references are appropriately cited.

Minor remarks:

1.       P.2, line 65-67 – this sentence is not clear to me. Without explanation how glycerol is connected to lipids it doesn’t make sense here.  

2.       P.2, line 70, “provide” should be changed to “providing”

3.       P.2, line 78 – in the Table 1 legend the phrase “are tabulated here” is not necessary

4.       Table 1 – please give the full names of the genes from column 1

5.       Table 1, it should be bbc1 instead of bcc1

6.       P.3, line 83 – “that improves” should be replaced by “and improve”

7.       P.3, line 91, there should be space between 750 and nm

8.       P.3, line 104, the phrase “in number” is not necessary

9.       P.3, line 105, please give the full names for WSC, and VSP4

10.   P.5, line 141, please remove the last sentence from the legend of Figure 1

11.   Figure 1 – G should be changed into E and E should be changed into G as the order is confusing to follow.

12.   In the legend for figures 2a-2c the word “image” should be in plural

13.   P.7, line 205, Picochlorum should be in italics

14.   P.8, line 214 – the last sentence should address where or for what the “other carbohydrates” are important  

15.   P.8, line 238, +2 should be in upper index

16.   P.9, line 273 – the word “were” is not necessary

17.   P.9, line 280 – last sentence should be rewritten (“to be studied” and “to study” are used in once sentence)

18.   P.9, line 295 – “transcription” should be changed into “transcriptomic”

19.   P.10, line 313 – the word “salt” should be added twice before “concentrations”

20.   P.11, line 341 – “an” is not necessary

21.   P.11, line 351, “-encoding gene” should be added to acetyl-CoA synthase

22.   P.11, line 366 – the word “achieved” sounds better than “selected for”

23.   P.12, line 395, “in” should be used after “resulting”

24.   P.12, line 399 – “to salt stress” should be added after “adaptation”

Author Response

Please find enclosed our answers to all your comments. Thank you for reviewing the manuscript. 

Round 2

Reviewer 1 Report

I still recommend changing “high salinity...” to “salt stress or deviation from optimal salinity...” in the first sentence of the abstract. It will be more correct.

Reviewer 3 Report

My principal remark will concern the main substance of this article that is clearly preliminary and need more investigation to assess a real review paper stage. An intensive work is still mandatory before envisaging to resubmit such an article.